**Data Availability Statement:** All relevant data are within the paper and its Supporting Information files.

**Funding:** This research is supported by the Digital Health CRC Limited (DHCRC) and is part of a larger

# Prediction models used in the progression of chronic kidney disease: A scoping review

**David K. E. Lim****[1]\***, **James H. Boyd[1,2]**, **Elizabeth Thomas[1,3]**, **Aron Chakera[3,4]**,
**Sawitchaya Tippaya[5]**, **Ashley Irish[6]**, **Justin Manuel[6]**, **Kim Betts[1]**, **Suzanne Robinson[1,7]**

**1** Curtin School of Population Health, Curtin University, Perth, WA, Australia, **2** La Trobe University, Melbourne, Bundoora, VIC, Australia, **3** Medical School, The University of Western Australia, Perth, WA, Australia, **4** Renal Unit, Sir Charles Gairdner Hospital, Perth, WA, Australia, **5** Curtin Institute for Computation, Curtin University, Perth, WA, Australia, **6** WA Country Health Service, Perth, WA, Australia, **7** Deakin Health Economics, Deakin University, Burwood, VIC, Australia

\* David.K.Lim@curtin.edu.au

## Abstract

### Objective

To provide a review of prediction models that have been used to measure clinical or pathological progression of chronic kidney disease (CKD).

### Design

Scoping review.

### Data sources

Medline, EMBASE, CINAHL and Scopus from the year 2011 to 17th February 2022.

### Study selection

All English written studies that are published in peer-reviewed journals in any country, that developed at least a statistical or computational model that predicted the risk of CKD progression.

### Data extraction

Eligible studies for full text review were assessed on the methods that were used to predict the progression of CKD. The type of information extracted included: the author(s), title of article, year of publication, study dates, study location, number of participants, study design, predicted outcomes, type of prediction model, prediction variables used, validation assessment, limitations and implications.

### Results

From 516 studies, 33 were included for full-text review. A qualitative analysis of the articles was compared following the extracted information. The study populations across the studies were heterogenous and data acquired by the studies were sourced from different levels and locations of healthcare systems. 31 studies implemented supervised models, and 2 studies

4-year collaborative partnership between Curtin University, La Trobe University, WA Department of Health, WA Country Health Services, WA Primary Health Alliance, and the DHCRC. The DHCRC is funded under the Commonwealth's Cooperative Research Centres (CRC) Program, project ID DHCRC-0073. The funders had no role in study design, data collection and analysis, decision to publish, or preparation of the manuscript.

**Competing interests:** The authors have declared that no competing interests exist.

included unsupervised models. Regardless of the model used, the predicted outcome included measurement of risk of progression towards end-stage kidney disease (ESKD) of related definitions, over given time intervals. However, there is a lack of reporting consistency on details of the development of their prediction models.

## Conclusions

Researchers are working towards producing an effective model to provide key insights into the progression of CKD. This review found that cox regression modelling was predominantly used among the small number of studies in the review. This made it difficult to perform a comparison between ML algorithms, more so when different validation methods were used in different cohort types. There needs to be increased investment in a more consistent and reproducible approach for future studies looking to develop risk prediction models for CKD progression.

## Introduction

Chronic Kidney Disease (CKD) is a global health burden with an estimated 5 to 10 million annual deaths worldwide due to kidney disease [1, 2]. Current data predict CKD will be the fifth leading cause of death worldwide by the year 2040 [3]. CKD is characterised by a gradual loss of the kidney's ability to remove wastes from the blood, and the severity of the disease is determined by the individual's estimated glomerular filtration rate (eGFR) [4]. CKD is arbitrarily categorised into five progressive stages with stage five often referred as end-stage kidney disease (ESKD), and its progression often leads to multiple overlapping complications [5, 6]. There is a spectrum of pathological, hereditary, and sociodemographic factors known to contribute to a decline in kidney function [5–11]. These factors include age ($\geq$60 years), smoking, low socioeconomic status, diabetes, hypertension, cardiovascular disease, body mass index ($\geq$30 kg/m$^2$), family history of kidney disease and use of pain-reliving medications [9–11].

The global nephrology community recognises that current models of care are insufficient to curb the growing CKD burden and that new care models are required to improve patient outcomes [12–14]. It has been suggested that the management framework for CKD needs to consider the disease across the entire life course of each individual [13]. New care models also need to consider improvements in areas such as disease surveillance, mitigation of risk factors, expanding research knowledge, and developing novel clinical interventions to slow the progression of CKD [13]. Despite having identified a number of risk factors associated with the onset of CKD, gaps remain in the methods for predicting the risk of CKD progression and interventions to slow CKD progression [13, 15, 16]. In addition, a large number of patients with CKD remain undetected through health systems [16] and clinicians have the challenge of managing the growing number of cases with limited tools for triaging patients.

### Predictive modelling techniques

Predictive modelling techniques applied to the growing number of clinical datasets have shown promise in accurately predicting the progression of chronic disease in the population [17–23]. Previous attempts have employed a wide range of prediction models, from well-established generalised linear models to more recent Machine Learning (ML) techniques [17–23]. Renal clinicians and researchers recognise the significant potential in developing risk

prediction models that can improve our ability to identify individuals at risk, in addition to potentially improving our understanding of the natural history of disease progression and contribute to the clinical management of CKD [22, 24, 25]. The application of ML models provides capacity to tap into the information contained in large and complex datasets and exploit the complex non-linear dependencies [18, 21, 23, 26–28]. The application of these analytical techniques promises to improve our understanding of CKD progression and inform key interventions to help slow progression and reduce the burden of CKD [11, 29–31]. Moreover, it can help inform clinicians with regards to treatment options by increasing confidence in the patient's likely prognostic course [32, 33].

Whilst the use of predictive modelling is gaining traction in CKD research, efforts are beset by the lack of a uniform approach to the reporting of important methodological advancements and developments of prediction models for CKD progression [23–25, 34]. This lack of consistent reporting of key characteristics and the evaluation of model performance has likely impeded uptake and support of prediction models by clinicians, while undermining reproducibility of research and clinical utility [24]. An example of a standardised reporting guidelines can be seen with the Equator Network who published the Transparent Reporting of a multivariable prediction model for Individual Prognosis Or Diagnosis (TRIPOD) statement that consists of a checklist considered vital by healthcare professionals, methodologists and journal editors, for the transparent reporting of multivariable prediction model studies [35, 36]. By implementing such a checklist, reporting can be standardised and reproducibility improved while facilitating progress towards cross-validation between different health settings and populations globally.

With inconsistency in the advancements of predictive modelling used in CKD progression analysis, this paper provides timely evidence from a scoping review about prediction models used in the progression of CKD. The review aims to 1) Identify and outline existing models used in predicting CKD progression; 2) To understand what measured outcome(s) and selected significant variables were chosen when building a prediction model for CKD progression. Its results will help inform clinical and scientific developments in this area and provide a better understanding of CKD progression.

## Classification of predictive models

Predictive modelling techniques can be generally classified into four broad categories; supervised, unsupervised, semi-supervised and reinforcement learning; with supervised and unsupervised being the most commonly applied in the medical field [18, 22, 25, 33]. This was also reflected in this scoping review where only supervised and unsupervised techniques were found in the studies that were assessed for full-text and will be discussed in later sections.

Supervised techniques can be further divided by the type of outcome they predict, with the two major groupings including continuous outcomes and categorical outcomes [37]. The regression technique is utilised when output variables are continuous data, such as values for weight or height [37]. On the other hand, classification techniques are commonly used for simpler data such as nominal or categorical data, where a simple binary outcome or a few predetermined categorical responses are required [37]. Supervised techniques have their own challenges and require sufficiently large volumes of correctly labelled data initially to perform accurately [26]. Some examples of commonly used supervised machine learning algorithms are linear or logistic regression, artificial neural networks, decision trees, k-nearest neighbours (KNN), random forest for classification, gradient boosting and support vector machines (SVM) [37, 38].

Unsupervised techniques can be further grouped into 2 types, clustering or association. Clustering is the process of segregating data into groups according to similar characteristics, whereas association is the process of identifying newer relationships within datasets based on certain selected attributes of the data. Additionally, unsupervised algorithms do not need manual labelling of datasets, as they can group data into clusters or identifying associations by themselves [26, 38]. The end result of these methods is to provide a simplified interpretation of a complex dataset, and often to sort observations into groups [38]. These groups can then be inspected for their ability to predict the outcome of interest. Some common examples of unsupervised ML algorithms include K-means clustering, mixture models, distribution models, dimensionality reduction, independent component analysis and principal component analysis.

## Methods

A scoping review was selected as it allows identification and mapping of existing evidence and to investigate and determine the knowledge gaps surrounding the topic [39]. This method is suitable for examining emerging evidence across a broad field of study and was guided by the PRISMA extension for Scoping Reviews (PRISMA-ScR), following a standardised approach to search, screen, and report articles [40].

### Data sources and searches

This scoping review was performed in the context of a larger study that investigates improving chronic kidney disease outcomes using linked routine records. With this context in mind, an initial concept grid was developed to address the objectives of the scoping review, together with the subsequent search histories that can be found in S1 Appendix. The review included studies in the past 10 years that developed or utilised any type of predictive modelling to predict the progression of CKD towards more severe stage of the disease. Articles included were published in peer-reviewed journals from any country, in the English language, between 1st January 2011 to 17th February 2022 inclusive. The which addresses the objectives of the scoping review. Four electronic databases, Medline, EMBASE, CINAHL, and Scopus were chosen for their bibliographic peer-reviewed publications that covers a broad range of medical life sciences, allied health, nursing and healthcare. Fig 1 illustrates the overall flow diagram of the literature review.

### Study selection and search

Four main overarching concepts, as described in the concept grid, were selected for the development of the search strategy, they were: *kidney disease*; *disease progression*; *techniques*; *outcomes*. The initial search strategy was developed for use in Medline and subsequently adapted for the other databases- keywords and sub headers were amended to reflect search terms used in each respective database. The steps used in Medline are as follows:

1. (chronic kidney disease* or chronic renal disease* or CKD or kidney disease* or kidney failure).ti,ab.

2. Renal Insufficiency, Chronic/ or Kidney Failure, Chronic/ or Diabetic Nephropathies/

3. 1 or 2

4. (progress* adj7 (CKD or disease)).ti,ab.

5. Disease Progression/

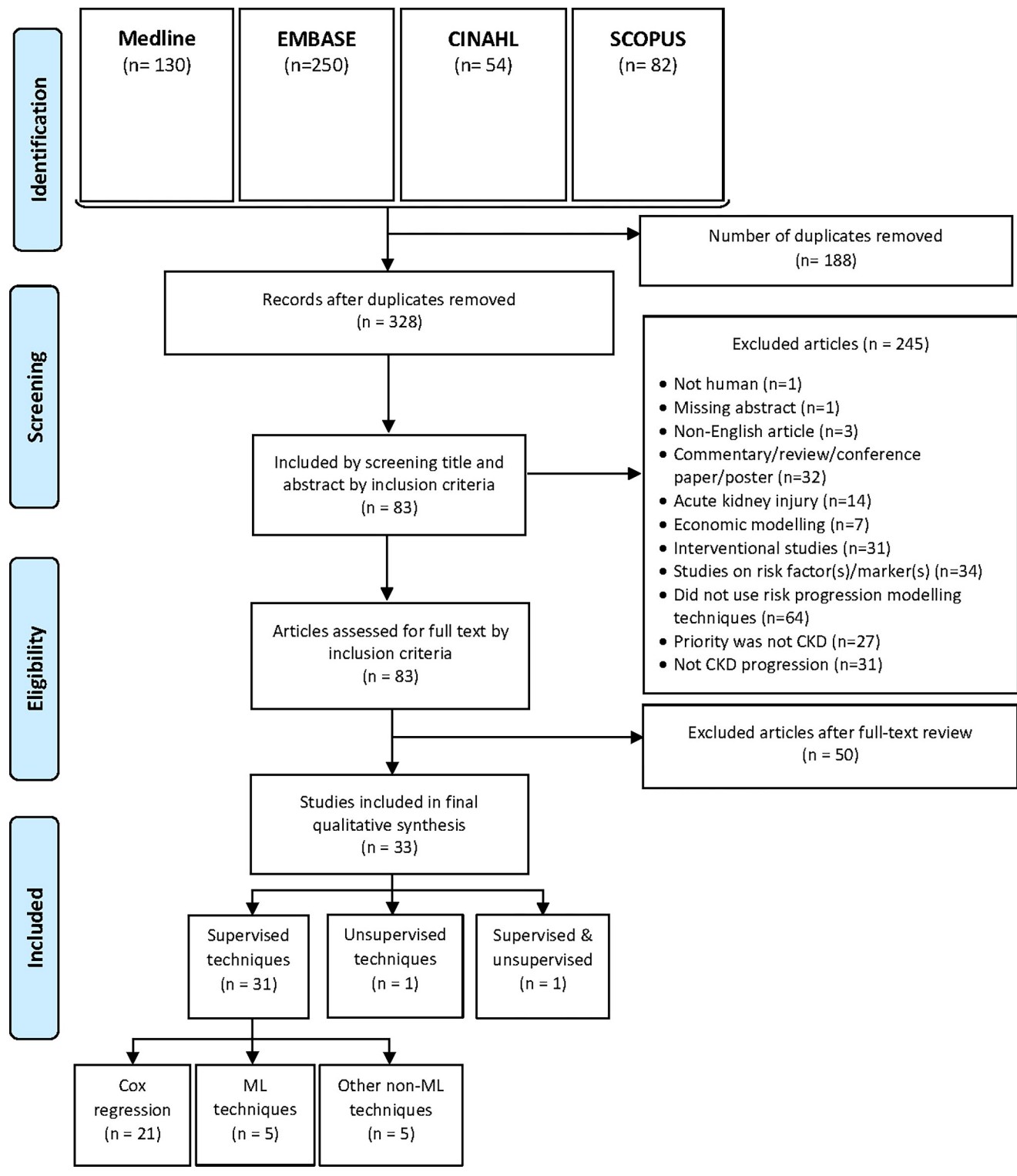

**Fig 1. PRISMA flow diagram.**

6. 4 or 5

7. (deep learning or machine learning or artificial intelligence or algorithms or prediction model* or statistic* model*).ti,ab.

8. Artificial Intelligence/ or Big data/ or machine learning/ or algorithms/ or models, statistical/

9. 7 or 8

10. (End stage renal disease or ESRD or Transplant* or Hemodialysis or Hospitali?ation or Mortality or Morbidity or Heart failure or Stroke).ti,ab.

11. Dialysis/ or Peritoneal Dialysis/ or Renal Dialysis/ or Kidney Transplantation/ or Cardio-vascular Diseases/ or Hypertension/ or Coronary Artery Disease/ or Coronary Disease/ or Hospitalization/ or Heart failure/ or Stroke/

12. 10 or 11

13. 3 and 6 and 9 and 12

14. limit 13 to (english language and yr = "2011 -Current")

The first key concept for *kidney disease* included keywords and MeSH terms used in steps 1 and 2, to capture different types of chronic kidney diseases, such as diabetic nephropathies or similar diseases, since it is chronic disease with multiple overlapping manifestations with associated comorbidities and risk factors [39]. The type of model used was not limited and included either statistical or ML algorithms used to predict CKD progression towards a wide range of clinical outcomes. A clear distinction was made that the study should examine prediction models for CKD progression, rather than models that predicted the onset of CKD.

## Title and abstract screening

All articles were exported into EndNote and duplicate articles were removed. Two independent reviewers performed title and abstract screening by applying inclusion and exclusion criteria. Studies included in the review were based on inclusion criteria, which included an implementation of a predictive model that was developed through analysis of health records; and they also had to include a reported outcome on the progression of CKD. The list of exclusion criteria can be found in Table 1.

The authors recognised that CKD is a very broad topic and did not place restrictions on the type of predictive model that was developed, the population of interest, the source of health data records, the predictive variables that were used, or a specific outcome. If there were any disagreements to the exclusion of articles, it was resolved through a discussion between the two reviewers—if required, a third reviewer for adjudication.

**Table 1. Exclusion criteria during title and abstract screening.**

| |
|---|
| • Animal studies |
| • Was not in the English language |
| • The study's primary focus was not on the progression of CKD |
| • The article was a commentary, conference paper, editorial, a review, an opinion piece, a supplementary abstract. |
| • The study did not consider determining progression of CKD from data records. |
| • Study that looked at risk factors, specific markers, case study |
| • Interventional studies |

## Data extraction and quality assessment

The researchers wanted to better understand the significant considerations taken into account when developing a prediction model for CKD progression, and to explore how these studies measured CKD progression [35]. The information extracted followed the items listed on the TRIPOD statement such as the article's title, author(s), publication year, year of study period, study locations and population size, study design (retrospective or prospective), predicted outcome(s), type of prediction model, predictors in the model, validation assessment, limitations, implications, eGFR formula and data balancing. Corresponding authors were contacted by email if full text was not available and were excluded if unobtainable.

## Results

The initial search had a combined total of 516 articles across Medline, EMBASE, CINAHL and Scopus, of which 188 duplicates were removed. 328 articles were then screened for their title and abstract, of which 245 articles were excluded based on exclusion criteria. 83 articles were then assessed for full-text eligibility by inclusion criteria, and subsequently 33 articles remained and were included in final qualitative review. Table 2 summarises the final articles that were included for full-text review.

### Predicted outcomes

It was generally observed that regardless of the model used, the predicted outcome included measurement of risk towards ESKD which were defined as [41, 43, 44, 48–50, 53, 56–60, 64, 69, 70, 72]:

1. when the eGFR value is <15 mL/kg/min/1.73 m$^2$ and / or

2. the initiation of kidney replacement therapies (KRTs) such as dialysis or kidney transplantation.

This risk of ESKD was generally predicted for specified time intervals of 1, 2, 3, and 5 years for supervised models, and shorter time intervals of 3, 6, 12 and 18 months for unsupervised models. There were very few studies that had predicted outcomes such as progression from an earlier stage to a more severe stage of CKD, for example from stage 1 to stages 3 or 4 [54, 56], and other predicted endpoints of stated percentage decline in eGFR levels [9, 41]. Depending on the quality of the available dataset [59, 60], the predicted outcome could also be combined with other variables such death, comorbidities, the type of dialysis and the time of diagnosis [46, 59, 68]. Some examples of outcomes that integrated these additional variables include, predicting the chances of future KRT at the time of CKD diagnosis [70]; a $\geq$ 50% decline in the eGFR from baseline [50] or an eGFR decline $\geq$30% from baseline [41]; the 5-year risk of KRT in CKD stage 3 and 4 [56]; the mortality and progression to ESKD over five years [65].

### Type of predictive model

Fig 1 shows that 31 studies implemented supervised models, and only 2 studies included unsupervised models with 1 of these 2 studies being a comparison study between supervised and unsupervised models. Of the studies that used supervised models, 21 studies implemented cox proportional hazards regression [41–61]. Seven studies used machine learning (ML) methods [9, 67–71], and one compared the performance among a number of ML techniques [70]. One study developed a model using Random Forest regression [68], and another study implemented a disease2disease model by first learning the International Classification of Diseases and then clustering the data into groups by considering the variables within the dataset [69]. A multistate marginal structural model (MS-MSM) was also developed in one study that

**Table 2. Summary of full-text review.**

| Author(s), Title of article | Year of publication (study dates) | Study location (n = size of cohort) | Study design (retrospective or prospective) | Predicted Outcome(s) | Type of prediction model | Predictors in the model | |
|---|---|---|---|---|---|---|---|
| | | | | | | Modifiable | Non-modifiable |
| **Supervised technique–cox regression** | | | | | | | |
| Akbari et al., Prediction of Progression in Polycystic Kidney Disease Using the Kidney Failure Risk Equation and Ultrasound Parameters [41] | 2020 (Jan 2010 –Jun 2017) | Eastern Ontario, Canada (n = 340) | Retrospective | A composite of 1) eGFR decline ≥30% from baseline and/or 2) the need for KRT (initiation of dialysis or pre-emptive transplantation). | Cox proportional hazards | Co-morbidities (cardiac disease, cancer, diabetes, hypertension, hyperlipidaemia), longitudinal biochemistry (proteinuria, eGFR by CKD-epi), kidney failure risk equation (KFRE), systolic blood pressure (SBP), and total kidney volume (TKV) were modelled as continuous predictors. | Age, sex. |
| Chang et al., A predictive model for progression of CKD [42] | 2019 (2006–2013) | Taiwan (n = 1549) | Retrospective | Kidney failure; dialysis. | Cox proportion hazard model survival analysis was used to investigate the risks of CKD progression to dialysis | Primary disease category, risk factors, co-morbidities (hypertension, hyperlipidaemia, hyperglycaemia, proteinuria, hypoproteinemia), and biochemical test values. | Age, sex, family medical history. |
| Cornec-Le Gall et al, The PROPKD Score: A New Algorithm to Predict Renal Survival in Autosomal Dominant Polycystic Kidney Disease [43] | 2016 (2009–2015) | Brittany, France (n = 1341) | Retrospective | PROPKD score: low, intermediate, and high risk for progression to ESKD. | Multivariate cox regression | Need for antihypertensive therapy before 35 years of age (referred hereinafter as age at hypertension onset, occurrence of the first urologic event before 35 years of age, and genetic status. | Age, sex. |
| Crnogorac et al., Clinical, serological and histological determinants of patient and renal outcome in ANCA-associated vasculitis (AAV) with renal involvement: an analysis from a referral centre [44] | 2017 (Jan 2003 –Dec 2013) | University Hospitals Dubrava and Merkur, Zagreb, Croatia (n = 83) | Retrospective | Primary outcome was combined endpoint patient death or progression to ESKD. Secondary outcomes were patient survival and progression to ESKD (kidney survival) singularly and disease relapse. | Univariate and multivariate cox proportional hazards regression analysis for each outcome was done. Multivariate Cox proportional hazards regression was done using backward stepwise analysis. Time to outcomes survival analysis was made using Kaplan–Meier estimates and categories were compared using log-rank test. | eGFR, Proteinuria, CRP, renal syndrome, pathohistological phenotype (normal/crescentic/sclerotic glomeruli, IFTA, fibrinoid necrosis) | Age, gender, time to diagnosis (months). |
| Dai et al., A predictive model for progression of chronic kidney disease to kidney failure using a large administrative claims database [45] | 2021 (2015–2017) | United States (n = 74,114) | Retrospective | From CKD stages 3 or 4 who were at high risk for progression to kidney failure | Logistic regression model. | CKD stage, hypertension (HTN), diabetes mellitus (DM), congestive heart failure, peripheral vascular disease, anaemia, hyperkalaemia (HK), prospective episode risk group score, and poor adherence to renin-angiotensin-aldosterone system inhibitors. The strongest predictors of progression to kidney failure were CKD stage (4 vs 3), HTN, DM, and HK. | Age, sex. |

*(Continued)*

**Table 2.** (Continued)

| Author(s), Title of article | Year of publication (study dates) | Study location (n = size of cohort) | Study design (retrospective or prospective) | Predicted Outcome(s) | Type of prediction model | Predictors in the model | |
|---|---|---|---|---|---|---|---|
| | | | | | | **Modifiable** | **Non-modifiable** |
| Dunkler et al., Risk Prediction for Early CKD in Type 2 Diabetes [46] | 2015 (2001–2008, 2003–2011) | The ONTARGET (n = 25,620) and the ORIGIN Trial (n = 12,537)–over 40 countries | Prospective | The outcome states after 5.5 years of follow-up were defined as alive without CKD, alive with CKD, or dead. | Two prediction models were developed: a laboratory model, containing laboratory markers of kidney function, sex and age, and a clinical model, containing the same markers and some clinical variables. Multinomial logistic regression was applied to develop prediction models for the three outcome states. | Baseline albuminuria, eGFR, UACR (urinary albumin-creatinine ratio), eGFR, albuminuria stage (normo- or microalbuminuria) | Age, sex. |
| Halbesma et al., Development and validation of a general population renal risk score [47] | 2011 (1997–1998) | City of Groningen, Netherlands (n = 6,809) | Prospective | A risk score identifies patients at risk for progressive CKD, | Backward logistic regression analysis | Hypertension, smoking, BMI, baseline eGFR and eGFR2, urea & electrolytes (U&E), C-reactive protein (CRP), SBP, plasma total cholesterol, glucose, triglycerides, urinary albumin exretion, and known HTN. | Age, sex, family history for CVD/CKD. |
| Hasegawa et al., Clinical prediction models for progression of chronic kidney disease to end stage kidney failure under pre-dialysis nephrology care: Results from the chronic kidney disease Japan cohort study [48] | 2018 (2007–2008) | CKD-JAC study —Japan (n = 2034) | Retrospective | ESKD onset, defined as the need for dialysis or pre-emptive kidney transplantation at 3 years | Cox proportional hazard regression | Physical examination findings, including body mass index (BMI) and systolic blood pressure (SBP); comorbid conditions (diabetes and hypertension), laboratory variables (eGFR, the urinary albumin-creatinine ratio (UACR), serum creatinine, serum sodium, serum albumin (ALB), haemoglobin (Hb), serum calcium, serum phosphorus, intact parathormone (iPTH), and FGF-23). | Age, sex. |
| Kang et al., An independent validation of the kidney failure risk equation in an Asian population [49] | 2020 (Jan 2001 –Dec 2016) | Korea (n = 38,905) | Retrospective | 2- and 5-year risk of ESKD | Cox proportional hazards models were fit using the variables included in each of the original equations, and baseline hazard was analysed. | eGFR, UACR, serum calcium, serum phosphorus, serum ALB, serum total CO2, diabetes mellitus, and hypertension, were obtained to calculate the KFREs. Because bicarbonate is not checked routinely, total $CO_2$ value was used as a bicarbonate value. | Age, sex. |
| Kataoka et al., Time series changes in pseudo-$R^2$ values regarding maximum glomerular diameter and the Oxford MEST-C score in patients with IgA nephropathy: A long-term follow-up study [50] | 2020 (1993–2017) | Kameda General Hospital, Japan (n = 43) | Prospective | Primary outcome was kidney disease progression, defined as ≥ 50% eGFR decline from baseline, or the development of ESKD requiring dialysis. | Kidney prognostic factors were also evaluated in cox regression analyses, and the Kaplan-Meier method was used for survival analyses. The prognostic variables for the kidney outcomes were assessed using univariate and multivariate cox proportional hazards models. | BMI, eGFR, laboratory results (urea and electrolytes, triglycerides, immunoglobulins, proteinuria), comorbidities, concomitant drugs, initial treatments, histological findings. | Age, sex. |

(Continued)

**Table 2.** (Continued)

| Author(s), Title of article | Year of publication (study dates) | Study location (n = size of cohort) | Study design (retrospective or prospective) | Predicted Outcome(s) | Type of prediction model | Predictors in the model | |
|---|---|---|---|---|---|---|---|
| | | | | | | Modifiable | Non-modifiable |
| Kim et al., Systolic blood pressure and chronic kidney disease progression in patients with primary glomerular disease [51] | 2021 (2005–2017) | Korea (n = 157) | Retrospective | A composite including ≥ 50% decrease in eGFR from the baseline (in at least two consecutive measurements), and ESKD (Initiation of maintenance dialysis or kidney transplantation). | A time-varying Cox model | BMI, smoking status, comorbid disease, glomerular disease type, laboratory measurements (eGFR, UPCR, total cholesterol, phosphorus, and ALB), medications (renin–angiotensin–aldosterone system (RAAS) blockers, diuretics, statins, immunosuppressive drugs), and remission status | Age, sex. |
| Li et al., Dynamic Prediction of Renal Failure Using Longitudinal Biomarkers in a Cohort Study of Chronic Kidney Disease [52] | 2017 | African American Study of Kidney Disease and Hypertension (AASK) (n = 1094) | Prospective | Survival regression models relating the predictor variables measured at or prior to the time of prediction to the time gap from the prediction time to the outcome event of interest (ESKD). | The Landmark Model and Predicted Probabilities. This is a variant of the Cox model. | Any hospitalization in the history window, the most recent log urine protein-to-creatinine ratio (Up/Cr) in the history window, the eGFR at the time of prediction, and the eGFR slope in the history window. | Age at the time of prediction |
| Maziarz et al, Homelessness and Risk of End-stage Renal Disease [53] | 2014 (Jan 1996 –Feb 2008) | Department of Public Health of the City and County of San Francisco (n = 16,656) | Retrospective | Risk of ESKD within 1, 3 and 5 years. | Linked with the national ESKD registry (United States Renal Data System) files based on patient last name, first name, date of birth, and Social Security Number. Four proportional hazards models each building on the previous, stratified by housing status. | eGFR, dipstick proteinuria, health insurance coverage, comorbidities (diabetes mellitus, CVD, hypertension, substance abuse, and chronic viral disease), and additional laboratory variables (serum ALB, serum calcium, serum cholesterol, and haemoglobin) | Age, sex, race-ethnicity |
| Palant et al., The association of serum creatinine variability and progression to CKD [54] | 2015 (1999–2005) | United States of America (n = 342,086) | Retrospective | Probability of entry into stage 4 CKD (30 mL/min/1.73 m$^2$) over a continuous timeline. | Logistic regression model. Time-to-event analysis was also used Kaplan-Meier and Cox regression | Initial eGFR, serum creatinine (SCr) variability, SCr slope, number of months with SCr readings, and comorbidities (DM, CAD, PNE, MI, angina, AKI, COPD, CHF). | Age, sex, race |
| Park et al., Predicted risk of renal replacement therapy at arteriovenous fistula referral in chronic kidney disease [55] | 2020 (May 2013 –May 2018) | Kaiser Permanente Northwest, Oregon and Washington, USA (n = 205) | Prospective | 2-year risk of KRT (following stage 4 CKD patients with 2-year observation period) | Cox regression model outlined by Schroeder et al. | eGFR (calculated from Chronic Kidney Disease Epidemiology Collaboration equation), haemoglobin, presence of proteinuria or albuminuria, systolic blood pressure, antihypertensive use, and Diabetes Complications Severity Index (The index was based on the International Statistical Classification of Diseases and Related Health Problems, Ninth Edition (ICD)-9 and, Tenth Edition 10 codes) | Age, sex |

*(Continued)*

**Table 2.** (Continued)

| Author(s), Title of article | Year of publication (study dates) | Study location (n = size of cohort) | Study design (retrospective or prospective) | Predicted Outcome(s) | Type of prediction model | Predictors in the model | |
|---|---|---|---|---|---|---|---|
| | | | | | | **Modifiable** | **Non-modifiable** |
| Schroeder et al., Predicting 5-year risk of RRT in stage 3 or 4 CKD: Development and external validation [56] | 2017 (Jan 2002 –Dec 2013) | Kaiser Permanente Northwest, USA (n = 22,460) | Retrospective cohort | Risk score for predicting the 5-year KRT risk for patients in stage 3 and 4 CKD. | A cox regression model using statistical methods described by Harrell and Steyerberg and endorsed by the Prognosis Research Strategy (PROGRESS) Group (26–28) and outlined in the TRIPOD guidelines. To avoid over-fitting the model, it required 20 KRT events per degree of freedom. | eGFR, hypertension, diabetes, and anaemia, proteinuria/albuminuria, body mass index (BMI), anti-hypertensive medication use, and prescription nonsteroidal anti-inflammatory drugs [NSAID] use. ICD-9 codes, counting complications such as: retinopathy, nephropathy, neuropathy, cerebrovascular disease, cardiovascular disease, peripheral vascular disease, and metabolic complications such as diabetic ketoacidosis. | Age, sex |
| Sun et al., Development and validation of a predictive model for end-stage renal disease risk in patients with diabetic nephropathy confirmed by renal biopsy [57] | 2020 (Feb 2012 –Dec 2018) | First Affiliated Hospital of Zhengzhou, China. (n = 968) | Retrospective | Primary outcome was a fatal or nonfatal ESKD event (peritoneal dialysis or haemodialysis for ESKD, kidney transplantation, or death due to chronic kidney failure or ESKD). ESKD was defined as 1) death due to diabetes with kidney manifestations or kidney failure; 2) hospitalization due to nonfatal kidney failure; and 3) an estimated GFR <15 mL/min/1.73 m$^2$ (National Kidney Foundation, 2002) | Multivariable logistic regression to identify baseline predictors for model development. | History of DM and HTN; laboratory parameters, including pathological grade (Class I, II a, II b, III, and IV represented as 1, 2, 3, 4, and 5 respectively), haemoglobin (Hb) levels, ALB levels, haemoglobin A1c (Hb$_{A1c}$) levels, blood urea nitrogen (BUN) levels, SCr levels, uric acid (UA) levels, cystatin C (CysC) levels, the estimated glomerular filtration rate (eGFR), 24-h urine protein levels, point total protein (TCr) levels, UACR, total cholesterol levels, triglyceride (TG) levels, HDL levels, LDL levels, serum lipid (HDL/total cholesterol ratio) levels; and inflammatory indicators such as PCT, ESR and CRP, creatine kinase isoenzyme (CKmb), B-type natriuretic peptide, and renin-angiotensin system blocker use. | Age, sex |
| Tangri et al., A Dynamic Predictive Model for Progression of CKD [58] | 2016 (Apr 2001—Dec 2009) | Outpatient CKD clinic of Sunnybrook Hospital in Toronto, Canada (n = 3004) | Prospective | Treated kidney failure, defined by initiation of dialysis therapy or kidney transplantation. | Cox proportional hazards models for time to kidney failure | Urinary albumin-creatinine ratio at baseline, eGFR, serum albumin, phosphorus, calcium, and bicarbonate values as time-dependent predictors. | Age, sex |

*(Continued)*

**Table 2.** (Continued)

| Author(s), Title of article | Year of publication (study dates) | Study location (n = size of cohort) | Study design (retrospective or prospective) | Predicted Outcome(s) | Type of prediction model | Predictors in the model | |
|---|---|---|---|---|---|---|---|
| | | | | | | **Modifiable** | **Non-modifiable** |
| Tangri et al, A predictive model for progression of chronic kidney disease to kidney failure [59] | 2011 (Apr 2001—Dec 2008) | Sunnybrook Hospital, Canada (n = 3449 and n = 4942) | Prospective | Risk categories (low, intermediate, high) of kidney failure at 1, 4, and 5 years— defined as initiation of dialysis or kidney transplantation and censored for mortality before kidney failure. Outcomes were ascertained by reviewing clinic records as well as through a matching algorithm with the Toronto Regional Dialysis Registry. Outcomes such as dialysis, death, and transplantation are all captured in the database, which matches all kidney failure outcomes with provincial and national registry | Developed sequentially using Cox proportional hazards regression methods. | Demographic variables, including; physical examination variables, including blood pressure and weight; comorbid conditions, including diabetes, hypertension, and aetiology of kidney disease; and laboratory variables from serum and urine collected at the initial nephrology visit. All predictor variables were obtained at baseline from the nephrology clinic EHR in the development data set | Age and sex |
| Xie et al., Risk prediction to inform surveillance of chronic kidney disease in the US Healthcare Safety Net: a cohort study [60] | 2016 (1996–2009) | Western United States (n = 28,779) | Retrospective | Risk of progression to ESKD (at years 1,3,5 and 7) and death, defined as having a first service date for maintenance dialysis or kidney transplantation. | Linkage to United States Renal Data System (USRDS). Calculated unadjusted incidence rates of ESKD for the full cohort, and for clinical subgroups defined by diabetes mellitus, hypertension, chronic viral diseases (HBV, HCV and/or HIV) and severe CKD (<30 mL/min/1.73m$^2$). We focused on these four subgroups because they represent common conditions frequently targeted by our Chronic Disease Management programs. Tested three proportional hazards regression models to predict progression to ESKD in each subgroup. | eGFR, dipstick proteinuria. | Age, sex, race |

*(Continued)*

**Table 2.** (Continued)

| Author(s), Title of article | Year of publication (study dates) | Study location (n = size of cohort) | Study design (retrospective or prospective) | Predicted Outcome(s) | Type of prediction model | Predictors in the model | |
|---|---|---|---|---|---|---|---|
| | | | | | | Modifiable | Non-modifiable |
| Xu et al., An easy-to-operate web-based calculator for predicting the progression of chronic kidney disease [61] | 2021 (Oct 2010 –Dec 2011) | Tokyo, Japan (n = 1,045) | Retrospective | 1-, 2-, and 3-year progression-free survival | Univariate and multiple Cox proportional hazard models | Aetiology (diabetes, nephrosclerosis, and Glomerulonephritis), haemoglobin level, creatinine level, proteinuria, and urinary protein/creatinine ratio | Age, sex |
| **Supervised technique–other non-ML** | | | | | | | |
| Diggle et al., Real-time monitoring of progression towards renal failure in primary care patients [62] | 2014 (Mar 1997—Mar 2007) | Salford Royal Hospital Foundation Trust (SRFT), Greater Manchester, UK (n = 22,910) | Retrospective | The predictive probability that they meet the clinical guideline for referral to secondary care. A person who is losing kidney function at a relative rate of at least 5% per year | The time-course of a person's underlying kidney function through a combination of explanatory variables, a random intercept and a continuous-time, non-stationary stochastic process. | eGFR, co-morbidities, and other baseline information. | Age, sex |
| Furlano et al., Autosomal Dominant Polycystic Kidney Disease: Clinical Assessment of Rapid Progression [63] | 2018 (Jan 2016 –Jun 2017) | Outpatient clinic in Spain (n = 305) | Retrospective | Rapid progression of disease according to their algorithm, including ultrasound, MRI measurements of kidney volume plus genetic testing historical eGFR. | ERA-EDTA WGIKD/ ERBP algorithm (European Renal Association-European Dialysis and Transplant Association (ERA—EDTA) Working Groups of Inherited Kidney Disorders and European Renal Best Practice (WGIKD/ ERBP), | Historical eGFR decline, historical TKV growth, age and height adjusted TKV, kidney length, PROPKD score, | Age, sex, family history |
| Lennartz et al., External Validation of the Kidney Failure Risk Equation and Re-Calibration with Addition of Ultrasound Parameters [64] | 2016 (CARE FOR HOMe study: 2008–2012—over 6 years & Hannover cohort: 1995–1999 | Saarland University hospital, Germany (n = 444) | Prospective | Risk of ESKD at 3 years following recruitment to validate KFRE | KFRE | eGFR (per 5 ml/min per 1.73 m$^2$, according to the MDRD formula), and urine albumin-to-creatinine ratio (ACR). eGFR and ACR were assessed as reported earlier. The KFRE prediction model formula with hazard ratios. | Age (per 10 years), sex |
| Nastasa et al., Risk prediction for death and end-stage renal disease does not parallel real-life trajectory of older patients with advanced chronic kidney disease-a Romanian center experience [65] | 2020 (Oct 2016—Oct 2018) | Romanian Outpatient Nephrology Department (n = 958) | Retrospective | Bansal score and KFRE give an estimate of mortality and progression to ESKD over five years | Individual risk for mortality was predicted using Bansal score, a nine-variable equation model developed in a US cohort of 828 participants aged ≥65 years with an eGFR For estimating the risk for progression to ESKD at 5 years, we used the 4-variable KFRE, according to the algorithm proposed by the ERBP guideline. | eGFR, clinical and biochemical variables | A set of demographic variables, not specific. |

*(Continued)*

**Table 2.** (Continued)

| Author(s), Title of article | Year of publication (study dates) | Study location (n = size of cohort) | Study design (retrospective or prospective) | Predicted Outcome(s) | Type of prediction model | Predictors in the model | |
|---|---|---|---|---|---|---|---|
| | | | | | | **Modifiable** | **Non-modifiable** |
| Zachasrias et al., A Novel Metabolic Signature To Predict the Requirement of Dialysis or Renal Transplantation in Patients with Chronic Kidney Disease [66] | 2019 (2010 – ongoing) | German Chronic Kidney Disease (GCKD) study (n = 4640) | Prospective | The Tangri score | Three proportional hazards models | eGFR, UACR, 24 NMR features (proton nuclear magnetic resonance (NMR) spectroscopy of blood plasma), creatinine, high-density lipoprotein, valine, acetyl groups of glycoproteins, and $Ca^{2+}$-EDTA carried the highest weights. | Age, sex. |
| **Supervised technique—ML** | | | | | | | |
| Cheng et al., Applying the Temporal Abstraction Technique to the Prediction of Chronic Kidney Disease Progression [9] | 2017 (Jan 2004 –Dec 2013) | Taiwan (n = 2066) | Retrospective | Predicting stage 4 CKD eGFR level decreasing to less than 15 ml/min/1.73 $m^2$ (ESKD) 6 months after collecting their final laboratory test information by evaluating time-related features | Several common supervised learning techniques, including C4.5, CART, and SVM. | TA-related variables (Temporal abstraction related variables), diabetes, blood pressure, drinking, smoking, heart disease, Variables exerting the greatest impact are consistent with those reported in previous studies, indicating that kidney function, BP, and blood haematocrit, were all vital indicators. | Age, sex. (Sex was the most critical factor affecting the deterioration of CKD among the first 25 variables that exerted the greatest impact.) |
| Makino et al., Artificial intelligence predicts the progression of diabetic kidney disease using big data machine learning [67] | 2019 (2005–2016) | Fujita Health University Hospital, Japan (n = 64,059) | Retrospective | Progression of type 2 diabetic kidney disease after 180 days (6 months) | Processing natural language and longitudinal data with big data machine learning. Applied logistic regression using the Python code with scikit-learn library for model solving. Among many machine learning packages including R, SPSS, Matlab, SAS, Weka and other, scikit-learn was chosen due to feature extraction processes written in Python. Due to the large number of explanation variables, L2-regularisation was used to avoid overfitting. | 36 features, where 12 sources (e.g., Urine protein, albuminuria and eGFR) were selectively chosen by extraction from known literature and 3 types of values (mean, latest, SD). | Past history of diseases. |
| Zhao et al., Predicting outcomes of chronic kidney disease from EMR data based on Random Forest Regression [68] | 2019 (2009–2017) | United States, Sioux Falls (n = 120,495) | Retrospective | The estimation of future eGFR value from the past eGFR values adjusted by clinical covariates, at year 1, 2 and 3. | Random Forest regression | eGFR, age, gender, body mass index (BMI), obesity, hypertension, and diabetes, which achieved a mean coefficient of determination of 0.95. | Age, sex. |

(*Continued*)

**Table 2.** (Continued)

| Author(s), Title of article | Year of publication (study dates) | Study location (n = size of cohort) | Study design (retrospective or prospective) | Predicted Outcome(s) | Type of prediction model | Predictors in the model | |
|---|---|---|---|---|---|---|---|
| | | | | | | **Modifiable** | **Non-modifiable** |
| Zhou et al., Use of disease embedding technique to predict the risk of progression to end-stage renal disease [69] | 2020 (Jan 2003 –Dec 2011) | California, United States (n = 35,844,800) | Retrospective | Progression of CKD to ESKD | Disease2disease (D2D) | Word2vec, comorbidities, ICD-9 or ICD-10 coding, five lab parameters: bicarbonate, calcium, protein, PTH, and urine protein/creatinine ratio, 25-OH vitamin, haematocrit, potassium, sodium and triglyceride. | Age, sex. |
| **Supervised and unsupervised techniques** | | | | | | | |
| Dovgan et al., Using machine learning models to predict the initiation of renal replacement therapy among chronic kidney disease patients [70] | 2020 (1998–2011) | Taiwan's national health insurance research database (NHIRD) (n = 23,948), | Retrospective | The onset of KRT at the time of CKD diagnosis—at 3, 6, and 12 months | Evaluated 10 ML algorithms that are implemented in the Python packages Scikit-learn and XGBboost: Decision Tree, Bagging Decision Trees, Random Forest, XGBoost, SVMs, Simple Gradient Descendent, Nearest Neighbours, Gaussian Naive Bayes, Logistic Regression, and Neural Network. Logistic Regression in combination with time features and data balancing, and without feature selection, filtering, or dimensionality reduction. | eGFR, albumin, haemoglobin, phosphorus, potassium, Correlations between diagnoses; diagnoses that are related to CKD, i.e., diabetes, HTN, hypertensive heart disease, glomerulonephritis, polycystic kidney, renal calculus, vesicoureteral reflux, kidney infections. | Age, sex, |
| Norouzi et al., Predicting Renal Failure Progression in Chronic Kidney Disease Using Integrated Intelligent Fuzzy Expert System [71] | 2016 (Oct 2002—Oct 2011) | Clinic of Nephrology, Imam Khomeini Hospital (Tehran, Iran) (n = 465) | Retrospective | Either GFR value less than 15 mL/kg/min/1.73 m$^2$, start of KRT or patient death, at 6, 12, or 18 months. | Adaptive neuro-fuzzy inference system (ANFIS) | Weight, underlying diseases, diastolic blood pressure, creatinine, calcium, phosphorus, uric acid, and GFR. | Age, sex, |

considers an estimated effect of time-dependent variables towards the predicted outcome [72]. Other ML algorithms that were tested include, neural networks, decision tree, random forest, XGBoost, Gaussian Naïve Bayes and logistic regression [57, 70, 71].

Three studies [49, 64, 65] performed an evaluation of the Kidney Failure Risk Equation (KFRE), and three other studies developed their own unique scoring algorithm that predicted ESKD [55, 63, 66].

## Significant variables in the model

Common predictors used in studies included age, sex, eGFR, urinary albumin to creatinine ratio (ACR), serum creatinine (SCr), diabetes, cardiovascular disease, body mass index (BMI),

and high blood pressure. Each predictive model was unique and incorporated different combinations of variables, and slightly different definitions of variables, such as high blood pressure. A recent paper by Xu et al. [61] published in 2021 highlighted that there are currently no robust biomarkers to predict progressive CKD, but rather relied on multiple longitudinal kidney measurements, such as eGFR and proteinuria.

The eGFR formula was also not consistent across studies, 13 studies used the CKD Epidemiology Collaboration (CKD-EPI) equation [9, 41, 42, 44, 46, 51, 54–56, 58, 63, 65, 66] and 9 studies used the Modification of Diet in Renal Disease (MDRD) equation [43, 45, 47, 49, 53, 60, 62, 64, 71]. Two studies used unique equations customised for their specific cohort [48, 69]. There were also 9 studies that did not specify the formula that they used to calculate the eGFR.

## Study population

The smallest study [50] had 43 participants and the largest study included over 300,000 patient records [54]. Included study populations were from the United States [45, 52–56, 60, 68, 69, 72], Canada [41, 46, 58, 59], Taiwan [9, 42, 70], Germany [64, 66], Japan [48, 50, 61, 67], France [43], Croatia [44], Korea [49, 51], United Kingdom [62], Iran [71], Romania [65], Spain [63], Netherlands [47] and China [57].

These studies that investigated on CKD progression used data records that were sourced from all levels of healthcare. Data records ranged from single medical facilities at a local level [58], to tertiary hospitals [50, 58, 59, 62, 67, 71], and to databases that were linked nationwide [42, 60, 69, 72]. The populations were also selected based on a particular comorbidity of interest, for example, polycystic kidney disease [41, 43, 63], ANCA associated vasculitis (AAV) [44], diabetes [46] or other cardiovascular conditions [53, 56, 67].

## Validation assessment

30 papers reported on the performance of their respective predictive models (regardless of the type of prediction model used) with 25 studies assessing the performance of their model by measuring the Area Under the Curve (AUC) [9, 41, 43, 45–54, 56–62, 64, 66–68, 70]. Both supervised and unsupervised techniques were shown to have used AUC to validate their prediction model, each having a relatively high value that indicated that their model was reliable in predicting their defined outcome. Relative performance of the prediction model was indicated using a variety of methods including sensitivity analysis, specificity, discrimination index and a goodness of fit analysis. However, only three studies were externally validated on an external population dataset [46, 56, 64].

Four studies explored the KFRE [41, 49, 64, 65] as a variable to try and improve the performance of their prediction model. Only one study reported using the F-score with confidence intervals [67], and there were a range of alternative measures that were used including the mean square error, mean absolute error, normalised mean square error, positive predictive values, negative predicted values, Harrell bootstrap resampling method, D-statistic and various confusion matrices [56, 68, 71].

## Missing data & imbalanced data

The most common limitation reported was missing or limited data, potentially due to the quality and availability of the data collected. Studies tried to overcome this issue by filling in the missing data using imputation techniques and internal validation techniques to help justify the dataset [42, 60]. There were also studies that reported having unbalanced data and outlined the methods applied to re-balance the data before initiating model development [9, 42, 45, 62, 70].

Knowing these limitations, studies recognised that their prediction models would only be applicable to their own given study population and would require external validation to allow generalisation of their model to other populations [41, 44–46, 49, 50, 53, 56, 59, 61, 64–66, 69, 73, 74].

## Discussion

The arrival of big data and data science techniques have supported better analytics using data from a variety of sources. However, many healthcare systems around the world are yet to fully utilise healthcare data for research purposes. Many of the data challenges within health relate to missing data, inconsistencies in recorded data and privacy concerns for linking data across organisations [75]. Despite these challenges, the application of health data is critical to support clinical decision making [31, 76, 77].

The success of disease management for conditions like CKD is dependent upon a clinician's ability to identify the risk of disease progression and poor outcomes. By utilising big data analytics, healthcare professionals may be able to predict disease progression in a timely manner, allowing the potential for better treatment for patients and reduced health costs.

Our review identified studies that had developed models to predict patient outcomes for CKD that measured the risk of progression towards ESKD over given time intervals. There was no single gold standard model identified, with each study producing its own unique prediction model, dependent on cohort's characteristics and quality of the available data. While Cox regression modelling was the predominant method; the burgeoning research on the use of ML techniques to improve the prediction of CKD progressing towards ESKD [23]. However, the decision to use a particular modelling technique should depend on finding the most suitable model based on the type of data available, size and dimensionality [19].

The application of both traditional and ML techniques have been explored as a way of determining the most significant variables or features for inclusion in the model [56, 70]. Studies that combined the use of both regression and ML techniques, first identified significant variables through regression prior to their inclusion into the development of a risk prediction model [56, 70, 78]. However, the practicality of determining significant features can be highly dependent on the availability and the quality of data. It is clear that the performance of a model is degraded if there is a lack of significant variables or if it includes irrelevant features [78–80]. Therefore, it is also recommended that future studies attempt to obtain whole population datasets that can help reduce the risk of missing data within the dataset and overcome the limitation of small study populations that are not generalisable to whole populations.

The study by Norouzi et al. demonstrated that an unsupervised adaptive neuro-fuzzy inference system (ANFIS), a type of neural network, was able to accurately predict GFR at sequential 6, 12 and 18-month intervals [71]. Other supervised non-ML models such as the KFRE and the ERBP algorithms, also produced results with high accuracy [63–65].

The comparison study by Dovgan et al. also demonstrated that features which correlated with a time approach produced the best results [70]. While the study did not include pathology results when developing their model, it produced the highest AUC via logistic regression, with XGBoost and Simple Gradient Descendent as a close second.

The MS-MSMs developed by Stephens-Shields et al. [72] developed a model that accounts for varying windows of time associated with different states while describing the effect of different exposures have on between states or endpoints. This is particularly applicable to the slow progression of CKD patients who enter the health system at different points in time and at various stages of the disease. In addition, each patient will have acquired different comorbidities and medical histories at different stages of their life.

Since the application of unsupervised and ML models are still in their exploratory stages, further research is required to investigate how these less explainable models manipulate very large and complex datasets that contain multi-dimensional and continuous variables [37, 78] and reflecting their application to predict CKD progression.

The review revealed a lack of consistent reporting of the methodology used for development and validation of prediction models. This often led to under reporting of model development, which hinders the ability of researchers to do a true comparison and externally validate their predictive models against existing models. This was emphasised when almost one third of studies reviewed did not report on the eGFR formula used, and is a significant limitation towards the development of this area of research. The development of a standardised reporting statement has yet to be widely implemented among CKD progression research which may be due to its relative novelty in the area of predictive modelling and statistical research [35].

Few studies explained how they attempted to re-balance their data, and methods differed for each study including log transformations, data resampling techniques, running simulation studies, and applying inversely proportional weights to class frequencies [9, 42, 62, 70]. The predictive models that have been developed are often difficult to implement locally as they lack information that allows clinicians to validate them. Limitations on data linkage within and between health organisations also contribute to the challenge of implementing this research, where siloed datasets are unlikely to be representative of whole populations. It is also recommended that future studies should include clear reporting of model development including any balancing of skewed datasets, steps to validate the model, and a description of how significant variables were chosen, which should theoretically at least include age, sex, eGFR (using a formula that provides reliable estimates for the study population), details on the population's characteristics, ACR, BMI and time-related variables if available.

A reliable risk prediction model for CKD progression would not only provide clinicians with earlier identification of CKD patients at greatest risk of progression, it would also enhance consultations and help clinicians determine suitable treatment options to improve patient outcomes [81, 82].

## Conclusions

Nephrology researchers are working towards producing an effective model to assist the detection of the risk of chronic kidney disease progression. The review highlights that supervised techniques, and more specifically, cox regression is the predominant model that is used to predict the progression of CKD. There were only a small number of studies in the review that used unsupervised and ML models, with the limited numbers making it very difficult to perform a comparison between these models. A more consistent and reproducible approach is required for future studies looking to develop risk prediction models for CKD progression. This would improve international collaborations and build upon the existing research to overcome the challenges to improve the effectiveness and reliability of these prediction models. Subsequently, this would also translate into enhanced health system planning, allocation of resources and improved health outcomes for CKD patients.

## Supporting information

**S1 Checklist. Preferred Reporting Items for Systematic reviews and Meta-Analyses extension for Scoping Reviews (PRISMA-ScR) checklist.**
(DOCX)

**S1 Appendix. Detailed search strategy per database.** Concept grids and search histories for Medline, EMBASE, CINAHL and Scopus.
(DOCX)

## Acknowledgments

This project is part of a larger 4-year collaborative partnership between Curtin University, La Trobe University, WA Department of Health, WA Country Health Service, WA Primary Health Alliance, and the DHCRC. All authors declare no conflict of interest and received no additional funding towards this manuscript. All authors also consent for publication and the supplementary data that supports the findings of this review will be available upon submission and publication.

## Author Contributions

**Conceptualization:** David K. E. Lim, Elizabeth Thomas, Suzanne Robinson.

**Data curation:** David K. E. Lim.

**Formal analysis:** David K. E. Lim.

**Funding acquisition:** Suzanne Robinson.

**Investigation:** David K. E. Lim, Elizabeth Thomas, Aron Chakera, Sawitchaya Tippaya, Suzanne Robinson.

**Methodology:** David K. E. Lim.

**Project administration:** David K. E. Lim.

**Resources:** David K. E. Lim.

**Supervision:** James H. Boyd, Elizabeth Thomas, Aron Chakera, Suzanne Robinson.

**Writing – original draft:** David K. E. Lim.

**Writing – review & editing:** David K. E. Lim, James H. Boyd, Elizabeth Thomas, Aron Chakera, Sawitchaya Tippaya, Ashley Irish, Justin Manuel, Kim Betts, Suzanne Robinson.

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
