## [Decision Letter · Decision Letter 0]

30 May 2022

PONE-D-22-10875Prediction Models Used in the Progression of Chronic Kidney Disease: A Scoping ReviewPLOS ONE

Dear Dr. Lim,

Thank you for submitting your manuscript to PLOS ONE. After careful consideration, we feel that it has merit but does not fully meet PLOS ONE’s publication criteria as it currently stands. Therefore, we invite you to submit a revised version of the manuscript that addresses the points raised during the review process.

We look forward to receiving your revised manuscript.

Kind regards,

Pierre Delanaye

Academic Editor

PLOS ONE

Journal Requirements:

"This project is supported by the DHCRC and is part of a larger 4-year collaborative partnership with Curtin University, La Trobe University, WA Department of Health, WA Country Health Service, WA Primary Health Alliance, and the DHCRC. All authors declare no conflict of interest and received no additional funding towards this manuscript. The DHCRC is funded under the Commonwealth's Cooperative Research Centres (CRC) Program (DHCRC-0073). All authors also consent for publication and the supplementary data that supports the findings of this review will be available upon submission and publication."

We note that you have provided funding information. However, funding information should not appear in the Acknowledgments section or other areas of your manuscript. We will only publish funding information present in the Funding Statement section of the online submission form. 

"This research is Digital Health CRC Limited (DHCRC) and is part of a larger 4-year collaborative partnership between Curtin University, La Trobe University, WA Department of Health, WA Country Health Services, WA Primary Health Alliance, and the DHCRC. The DHCRC is funded under the Commonwealth's Cooperative Research Centres (CRC) Program, project ID DHCRC-0073. The funders had no role in study design, data collection and analysis, decision to publish, or preparation of the manuscript."

Additional Editor Comments:

The authors should better justify the way some studies have been included or not. They should also better define outcomes and variables included in the models.

Reviewers' comments:

Reviewer's Responses to Questions

**Comments to the Author**

1. Is the manuscript technically sound, and do the data support the conclusions?

Reviewer #1: Yes

Reviewer #2: Partly

2. Has the statistical analysis been performed appropriately and rigorously? 

Reviewer #1: Yes

Reviewer #2: N/A

3. Have the authors made all data underlying the findings in their manuscript fully available?

Reviewer #1: Yes

Reviewer #2: Yes

4. Is the manuscript presented in an intelligible fashion and written in standard English?

Reviewer #1: Yes

Reviewer #2: Yes

5. Review Comments to the Author

Reviewer #1: Thank you for giving me the opportunity to review the manuscript by David Lim et al entitled: “Prediction models used in the progression of chronic kidney disease: a scoping review”.

In this manuscript, the authors propose a review of the different models that have been proposed in order to predict the risk of progression of CKD. This subject is of great importance, as the authors explain it in their introduction, from a clinician point of view but also from a public health point of view.

The manuscript is well written and easy to read. The review was built following the PRISMA statement, which is a pledge of quality.

A bullet point from this review is the lack of consistency regarding the definitions of the outcomes, the definitions of the variables included in the models

I only have few comments:

- I find it useful to have described the four categories of modelling techniques in the introduction.

- I would have liked to get more insight into the explanatory variables included in the different models, separating the modifiable variables from the non-modifiable variables.

- Could the authors add some details in the text about the different source populations. Some studies tried to explore the risk of progression of CKD in the general population, whereas in some others studies, it was in very specific populations: ANCA vasculitis, ADPKD…

- The last proposal of the results part: “Additionally, a more precise estimation of …. versus end-of-life care” is a different subject, so I would propose to remove this sentence and to insist on the lines 19 to 23 : “Future studies should… variables if available” which is the really important point nephrologists should focus on, based on the results of this review.

- Thank you for table S2!

- Given the aim of the article, to review the models used in the progression of CKD, could the authors add in the discussion part some commentaries about the best adapted models for further works, according to the variables used or the population studied?

Reviewer #2: Dear Editorial Board

David KE Lim et al present a scoping review about Prediction modelling on CKD progression.

A major comments is about context method and selection of article.

These is already some recent review publication about this topic. The authors needs to refer to these articles in introduction and justify why they don't find the same articles. If they want to study prediction model of CKD progression ESRD must be an outcome in research equation.

An important score of progression is the prediction model of Grams et al 2018 but was not cited in references. 2 authors did review on this topic and found several article not found in this scoping review ( Ramspeck CL & Prouvot J).

Grams ME, Sang Y, Ballew SH, Carrero JJ, Djurdjev O, Heerspink HJL, Ho K, Ito S, Marks A,

Naimark D, Nash DM, Navaneethan SD, Sarnak M, Stengel B, Visseren FLJ, Wang AY-M, Köttgen

A, Levey AS, Woodward M, Eckardt K-U, Hemmelgarn B, Coresh J: Predicting timing of clinical

outcomes in patients with chronic kidney disease and severely decreased glomerular filtration

rate. Kidney International 93: 1442–1451, 2018

Ramspek CL, de Jong Y, Dekker FW, van Diepen M: Towards the best kidney failure prediction

tool: a systematic review and selection aid. Nephrol Dial Transplant [Internet] Available from:

https://academic.oup.com/ndt/advance-article/doi/10.1093/ndt/gfz018/5369192

Low performance of prognostic tools for predicting dialysis in elderly people with advanced CKD.

Prouvot J, Pambrun E, Couchoud C, Vigneau C, Roche S, Allot V, Potier J, Francois M, Babici D, Prelipcean C, Moranne O; PSPA investigators. J Nephrol. 2021 Aug;34(4):1201-1213. doi: 10.1007/s40620-020-00919-6.

6. PLOS authors have the option to publish the peer review history of their article (what does this mean?). If published, this will include your full peer review and any attached files.

Reviewer #1: **Yes: **Antoine Lanot

Reviewer #2: No

---

## [Author Response · Author response to Decision Letter 0]

28 Jun 2022

Journal Requirements:

https://journals.plos.org/plosone/s/file?i d=wjVg/PLOSOne_formatting_sample_main_body.pdf and https://journals.plos.org/plosone/s/file?id=ba62/PLOSOne_formatting_sample_title_authors_affiliations.pdf

Response to reviewer: 

Please find the revised manuscript with all the necessary changes made to meet PLOS ONE’s style requirements, including file naming rules.

Response to reviewer:

We have now ensured that the grant award number matches with the funding information section, it should read that the Award Number: DHCRC-0073, by the Digital Health CRC Ltd.

"This project is supported by the DHCRC and is part of a larger 4-year collaborative partnership with Curtin University, La Trobe University, WA Department of Health, WA Country Health Service, WA Primary Health Alliance, and the DHCRC. All authors declare no conflict of interest and received no additional funding towards this manuscript. The DHCRC is funded under the Commonwealth's Cooperative Research Centres (CRC) Program (DHCRC-0073). All authors also consent for publication and the supplementary data that supports the findings of this review will be available upon submission and publication."

We note that you have provided funding information. However, funding information should not appear in the Acknowledgments section or other areas of your manuscript. We will only publish funding information present in the Funding Statement section of the online submission form. 

"This research is Digital Health CRC Limited (DHCRC) and is part of a larger 4-year collaborative partnership between Curtin University, La Trobe University, WA Department of Health, WA Country Health Services, WA Primary Health Alliance, and the DHCRC. The DHCRC is funded under the Commonwealth's Cooperative Research Centres (CRC) Program, project ID DHCRC-0073. The funders had no role in study design, data collection and analysis, decision to publish, or preparation of the manuscript."

Response to reviewer:

We have removed funding information from the Acknowledgements section in the manuscript, and it now reads:

“This project is supported by the DHCRC and is part of a larger 4-year collaborative partnership with Curtin University, La Trobe University, WA Department of Health, WA Country Health Service, WA Primary Health Alliance, and the DHCRC. All authors declare no conflict of interest and received no additional funding towards this manuscript. All authors also consent for publication and the supplementary data that supports the findings of this review will be available upon submission and publication.”

Response to reviewer:

We have added the “Supporting information captions” section on the last page of the manuscript:

“Supporting information captions

S1 Appendix. Detailed search strategy per database. The concept grid and search history for Medline, EMBASE, CINAHL and Scopus.”

Tables 1, 2, and Fig 1 have all been added appropriately into the manuscript body, they are no longer “supporting information”.

Additional Editor Comments:

The authors should better justify the way some studies have been included or not. They should also better define outcomes and variables included in the models.

Response to reviewer:

Thanks for the comment. This scoping review was done in context of a larger project that looks at improving chronic kidney disease outcomes through the use of linked routine pathology records and hospital records. With that context in mind, one of the project’s future objective is to explore how other studies have tried to understand the progression of CKD. The four main areas that would cover this future objective are as described in the concept grid found in S1 Appendix:

1) chronic kidney disease; and

2) Disease progression; and

3) different statistical / machine modelling techniques that were developed/used to understand the progression of CKD; and

4) A range of outcomes that are commonly associated with ESRD, with ESRD included as one of several other outcomes of the search strategy.

Additional clarity has been added to the method section:

“Methods

A scoping review was selected as it allows identification and mapping of existing evidence and to investigate and determine the knowledge gaps surrounding the topic [39]. This method is suitable for examining emerging evidence across a broad field of study and was guided by the PRISMA extension for Scoping Reviews (PRISMA-ScR), following a standardised approach to search, screen, and report articles. 

Data Sources and Searches

This scoping review was performed in the context of a larger study that investigates improving chronic kidney disease outcomes using linked routine records. With this context in mind, an initial concept grid was developed to address the objectives of the scoping review, together with the subsequent search histories that can be found in S1 Appendix.”

Outcomes from each study have been included in Table 2’s column “Predicted Outcome(s)”, with the types of variables distinguished as modifiable and non-modifiable.

Reviewers' comments:

Reviewer's Responses to Questions

Comments to the Author

1. Is the manuscript technically sound, and do the data support the conclusions?

Reviewer #1: Yes

Reviewer #2: Partly

Response to reviewer: Not required.

2. Has the statistical analysis been performed appropriately and rigorously? 

Reviewer #1: Yes

Reviewer #2: N/A

Response to reviewer: Not required.

3. Have the authors made all data underlying the findings in their manuscript fully available?

Reviewer #1: Yes

Reviewer #2: Yes

Response to reviewer: Not required.

4. Is the manuscript presented in an intelligible fashion and written in standard English?

Reviewer #1: Yes

Reviewer #2: Yes

Response to reviewer: Not required.

5. Review Comments to the Author

Reviewer #1: Thank you for giving me the opportunity to review the manuscript by David Lim et al entitled: “Prediction models used in the progression of chronic kidney disease: a scoping review”.

In this manuscript, the authors propose a review of the different models that have been proposed in order to predict the risk of progression of CKD. This subject is of great importance, as the authors explain it in their introduction, from a clinician point of view but also from a public health point of view.

The manuscript is well written and easy to read. The review was built following the PRISMA statement, which is a pledge of quality.

A bullet point from this review is the lack of consistency regarding the definitions of the outcomes, the definitions of the variables included in the models

I only have few comments:

*** I find it useful to have described the four categories of modelling techniques in the introduction.

Response to reviewer:

Thanks for letting us know that you found it useful, and have provided more clarity to have the description of the categories of the modelling techniques in the introduction under the subheading “Classification of predictive models”:

“Classification of predictive models 

Predictive modelling techniques can be generally classified into four broad categories; supervised, unsupervised, semi-supervised and reinforcement learning; with supervised and unsupervised being the most commonly applied in the medical field. This was also reflected in this scoping review where only supervised and unsupervised techniques were found in the studies that were assessed for full-text and will be discussed later in later sections.”

We have also provided more clarity when describing these categories of modelling techniques within Results section by adding a subheading “Type of predictive model”:

“Type of predictive model

Fig 1 shows that 31 studies implemented supervised models, and only 2 studies included unsupervised models with 1 of these 2 studies being a comparison study between supervised and unsupervised models….”

*** I would have liked to get more insight into the explanatory variables included in the different models, separating the modifiable variables from the non-modifiable variables.

Response to reviewer:

We have now included in Table 2 an extra split in the column for “Predictors in the model”, this has been done by further dividing this column into two columns namely “modifiable” and “non-modifiable”. 

*** Could the authors add some details in the text about the different source populations. Some studies tried to explore the risk of progression of CKD in the general population, whereas in some others studies, it was in very specific populations: ANCA vasculitis, ADPKD…

Response to reviewer:

We have included additional clarification about the different population sources within the results section, under “Study population”, which now reads:

“Study population

The smallest study had 43 participants and the largest study included over 300,000 patient records. Included study populations were from the United States, Canada, Taiwan, Germany, Japan, France, Croatia, Korea, United Kingdom, Iran, Romania, Spain, Netherlands and China. 

These studies that investigated on CKD progression used data records that were sourced from all levels of healthcare. Data records ranged from single medical facilities at a local level, to tertiary hospitals, and to databases that were linked nationwide. The populations were also selected based on a particular comorbidity of interest, for example, polycystic kidney disease, ANCA associated vasculitis (AAV), diabetes or other cardiovascular conditions.”

*** The last proposal of the results part: “Additionally, a more precise estimation of …. versus end-of-life care” is a different subject, so I would propose to remove this sentence and to insist on the lines 19 to 23 : “Future studies should… variables if available” which is the really important point nephrologists should focus on, based on the results of this review.

Response to reviewer: 

Thanks for the suggestion, we have removed the sentence, “Additionally, a more precise estimation of a patient’s risk of death after starting dialysis would also enable clinicians to better assist patients to consider the benefits of treatment versus end-of-life care”.

*** Thank you for table S2!

Response to reviewer: Not required.

*** Given the aim of the article, to review the models used in the progression of CKD, could the authors add in the discussion part some commentaries about the best adapted models for further works, according to the variables used or the population studied?

Response to reviewer:

We have added in more clarity within the discussion section which highlights the importance of trying to capture the whole population with a standardised reporting method, which improves the modelling process:

“The application of both traditional and ML techniques have been explored as a way of determining the most significant variables or features for inclusion in the model [56, 70]. Studies that combined the use of both regression and ML techniques, first identified significant variables through regression prior to their inclusion into the development of a risk prediction model [56, 70, 78]. However, the practicality of determining significant features can be highly dependent on the availability and the quality of data. It is clear that the performance of a model is degraded if there is a lack of significant variables or if it includes irrelevant features [78-80]. Therefore, it is also recommended that future studies attempt to obtain whole population datasets that can help reduce the risk of missing data within the dataset and overcome the limitation of small study populations that are not generalisable to whole populations…

… Limitations on data linkage within and between health organisations also contribute to the challenge of implementing this research, where siloed datasets are unlikely to be representative of whole populations. It is also recommended that future studies should include clear reporting of model development including any balancing of skewed datasets, steps to validate the model, and a description of how significant variables were chosen, which should theoretically at least include age, sex, eGFR (using a formula that provides reliable estimates for the study population), details on the population’s characteristics, ACR, BMI and time-related variables if available.”

Reviewer #2: Dear Editorial Board

David KE Lim et al present a scoping review about Prediction modelling on CKD progression.

A major comments is about context method and selection of article.

These is already some recent review publication about this topic. The authors needs to refer to these articles in introduction and justify why they don't find the same articles. If they want to study prediction model of CKD progression ESRD must be an outcome in research equation.

An important score of progression is the prediction model of Grams et al 2018 but was not cited in references. 2 authors did review on this topic and found several article not found in this scoping review ( Ramspeck CL & Prouvot J).

Grams ME, Sang Y, Ballew SH, Carrero JJ, Djurdjev O, Heerspink HJL, Ho K, Ito S, Marks A,

Naimark D, Nash DM, Navaneethan SD, Sarnak M, Stengel B, Visseren FLJ, Wang AY-M, Köttgen

A, Levey AS, Woodward M, Eckardt K-U, Hemmelgarn B, Coresh J: Predicting timing of clinical

outcomes in patients with chronic kidney disease and severely decreased glomerular filtration

rate. Kidney International 93: 1442–1451, 2018 (https://www.ncbi.nlm.nih.gov/pmc/articles/PMC5967981/)

Ramspek CL, de Jong Y, Dekker FW, van Diepen M: Towards the best kidney failure prediction

tool: a systematic review and selection aid. Nephrol Dial Transplant [Internet] Available from:

https://academic.oup.com/ndt/advance-article/doi/10.1093/ndt/gfz018/5369192

Low performance of prognostic tools for predicting dialysis in elderly people with advanced CKD.

Prouvot J, Pambrun E, Couchoud C, Vigneau C, Roche S, Allot V, Potier J, Francois M, Babici D, Prelipcean C, Moranne O; PSPA investigators. J Nephrol. 2021 Aug;34(4):1201-1213. doi: 10.1007/s40620-020-00919-6.

Response to reviewer:

Thanks for the comment. This scoping review was done in context of a larger project that looks at improving chronic kidney disease outcomes through the use of linked routine pathology records and hospital records. With that context in mind, one of the project’s future objective is to explore how other studies have tried to understand the progression of CKD. The four main areas that would cover this future objective are as described in the concept grid found in S1 Appendix, through the inclusion of keywords and subject terms with the help of the University’s Health Sciences faculty librarian:

1) chronic kidney disease; and

2) Disease progression; and

3) different statistical / machine modelling techniques that were developed/used to understand the progression of CKD; and

4) A range of outcomes that are commonly associated with ESRD, with ESRD included as one of several other outcomes of the search strategy.

After reviewing the Medline search strategy as described in S1 Appendix, page 2, it was found that the papers by Grams et al. and Prouvot et al., were excluded at step 9 of the search history. This meant that these papers did not include the listed keywords or MeSH terms of modelling techniques found in the concept grid column “Techniques” within their title or abstract. 

For the study by Ramspek et al., while this paper included kidney failure and prediction of disease progression, it does not report any specific outcomes in its title or abstract. More specifically, it was excluded at step 12 of the search history, which meant that the paper did not include the listed keywords or MeSH terms of any of the range of outcomes that are commonly associated with ESRD, within its title or abstract. 

6. PLOS authors have the option to publish the peer review history of their article (what does this mean?). If published, this will include your full peer review and any attached files.

Do you want your identity to be public for this peer review? For information about this choice, including consent withdrawal, please see our Privacy Policy.

Reviewer #1: Yes: Antoine Lanot

Reviewer #2: No

Response to reviewer: Not required.

---

## [Editor Report · Decision Letter 1]

5 Jul 2022

Prediction Models Used in the Progression of Chronic Kidney Disease: A Scoping Review

PONE-D-22-10875R1

Dear Dr. Lim,

We’re pleased to inform you that your manuscript has been judged scientifically suitable for publication and will be formally accepted for publication once it meets all outstanding technical requirements.

Kind regards,

Pierre Delanaye

Academic Editor

PLOS ONE

Additional Editor Comments (optional):

No further comments.
---

## [Editor Report · Acceptance letter]

15 Jul 2022

PONE-D-22-10875R1 

Prediction Models Used in the Progression of Chronic Kidney Disease: A Scoping Review 

Dear Dr. Lim:

I'm pleased to inform you that your manuscript has been deemed suitable for publication in PLOS ONE. Congratulations! Your manuscript is now with our production department. 

Kind regards, 

on behalf of

Professor Pierre Delanaye 

Academic Editor

PLOS ONE